

# Evaluation of surface shortwave downward radiation forecasts by the numerical weather prediction model AROME

Marie-Adèle Magnaldo[1], Quentin Libois[1], Sébastien Riette[1], and Christine Lac[1]

[1]Université de Toulouse, Météo-France, CNRS, Toulouse, France

**Correspondence:** Quentin Libois (quentin.libois@meteo.fr)

**Abstract.**

With the worlwide development of the solar energy sector, the need for reliable surface shortwave downward radiation (SWD) forecasts has significantly increased in recent years. SWD forecasts of a few hours to a few days based on numerical weather prediction (NWP) models are essential to facilitate the incorporation of solar energy into the electric grid and ensure
network stability. However, errors in NWP models can be substantial. In order to characterize in detail the performances of AROME, the operational NWP model of the French weather service Météo-France, a full year of hourly AROME forecasts is compared to corresponding *in situ* SWD measurements from 168 high-quality pyranometers covering France. In addition, to classify cloud scenes at high temporal frequency and over the whole territory, cloud products derived from the Satellite Application Facility for Nowcasting and Very Short Range Forecasting (SAF NWC) from geostationary satellites are also
used. The 2020 bias is 18 W m$^{-2}$ and the root-mean-square-error is 98 W m$^{-2}$. The situations that contribute the most to the bias correspond to cloudy skies in the model and in the observations, situations that are very frequent (66 %) and characterized by an annual bias of 24 W m$^{-2}$. Part of this positive bias probably comes from an underestimation of cloud fraction in AROME, although this is not fully addressed in this study due to lack of consistent observations at kilometer resolution. The other situations have less impact on SWD errors. Missed cloudy situations and erroneously predicted clouds, which correspond
on average to clouds with a low impact on the SWD, also have low occurrence (4 % and 11 %). Likewise, well-predicted clear sky conditions are characterized by a low bias (3 W m$^{-2}$). When limited to overcast situations in the model, the bias in cloudy skies is small (1 W m$^{-2}$) but results from large compensating errors. Indeed, further investigations show that high clouds are systematically associated with a SWD positive bias while low clouds are associated with a negative bias. This detailed analysis shows that the errors result from a combination of incorrect cloud optical properties and cloud fraction errors,
highlighting the need for a more detailed evaluation of cloud properties. This study also provides valuable insights into the potential improvement of AROME physical parametrization.

## 1  Introduction

In the context of global warming, the European Union's green transition plan calls for at least 45 % of the energy to come from renewable energy sources (RES) by 2030. This is expected to reduce greenhouse gas emissions by 55% compared to 1990 levels
(https://energy.ec.europa.eu/topics/renewable-energy/renewable-energy-directive-targets-and-rules/renewable-energy-targets_



en, last access: 25 April 2023). The share of RES has already increased significantly in Europe and worldwide over the last decade (International Energy Agency, 2019). In France, in 2021, 13 067 MWp (MW peak) of solar photovoltaic (PV) generation capacity was installed (which corresponds to the potential production under standard test conditions) and 2 687 MWp was added in 2022 (Réseau de transport d'électricité, 2021). Solar energy thus covers 3.1 % of the French annual electricity consumption (https://bilan-electrique-2021.rte-france.com/production_solaire/, last access: 25 April 2023). This highlights that solar energy is a key element in moving to a more sustainable energy system.

However, solar energy production is highly dependent on weather conditions, especially clouds, which are responsible for its very high spatio-temporal variability (Widén et al., 2015; Antonanzas et al., 2016). This variability is an issue for the planning of solar energy production and for the overall stability of the electric grid, which requires a balance between supply and demand (Antonanzas et al., 2016; Betti et al., 2021). To deal with that, accurate forecasts, mainly for surface shortwave downward radiation (SWD), are essential. Different forecasting horizons are relevant for the solar energy sector, ranging from minutes to months ahead (Das et al., 2018; Antonanzas et al., 2016; Raza, 2016; Betti et al., 2021). Very short term forecasts, covering horizons from seconds to one hour, are important for power smoothing, real-time electricity dispatch and optimal reserves optimizing storage. Short-term forecasts, from one hour to several days, are necessary for unit commitment, scheduling and dispatch of electrical power, but also to enhance the security of grid operation. Medium-term forecasts, from one week to one month, are useful, for example, for scheduling maintenance. Long-term forecasts, from one month to one year, are useful for planning electricity generation, transmission, and distribution aside from energy bidding and securing operation (Das et al., 2018). In a much longer term, in particular in a context of global warming, estimating the future potential of variable renewable resources, including solar, and the impact of temperature on electricity demand, are essential to provide an assessment of future energy systems (Dubus et al., 2022).

Depending on the time horizon, different forecasting techniques are used (Widén et al., 2015; Raza, 2016; Das et al., 2018; Betti et al., 2021). For instance, for infra-hour forecasts, the use of two or more all-sky imagers (ASI) organized in a network has proven successful (Nouri et al., 2019b, a; Logothetis et al., 2022; Chu et al., 2022). Forecasting techniques based on historical data, including machine learning approaches, can also be effective for the very short-term (Das et al., 2018). For short-terme forecast up to 6 h horizon, Satellite-based methods that extrapolate cloud locations using cloud-motion vectors can be effective (Cros et al., 2020, e.g.,). From hours to a few days, physical methods based on weather forecasts produced by Numerical Weather Prediction (NWP) models are the most common (Widén et al., 2015; Antonanzas et al., 2016; Betti et al., 2021). As a consequence, these NWP models, which solve the mesoscale atmospheric dynamics and account for small-scale physical processes, are an essential element for the management of power systems involving a significant amount of solar energy.

However, the performance of NWP models in predicting SWD remains limited. As an illustration, we compared the 1-day SWD forecasts of AROME (Seity et al., 2011), the operational NWP model of the French weather service Météo-France, to hourly averages from the national pyranometer network comprising 168 stations. For 2020, the mean annual bias and root mean square error (RMSE) are respectively 18 W m$^{-2}$ and 97 W m$^{-2}$, for a mean SWD of 340 W m$^{-2}$. This shows that the errors can be significant, with correspondingly high uncertainties in the PV production forecasts.

Nevertheless, until now, SWD errors have not been a priority for weather forecasts produced by NWP models, since they did



not have radiative scores, whereas they have been particularly studied in climate models. Growing interest from various end-users, including the PV community, is now highlighting these substantial SWD errors and the need for better forecasts.

For example, Nielsen and Gleeson (2018) evaluated the SWD forecasts of the NWP model HARMONIE-AROME in Denmark with a pyranometer network and found a negative bias for days with optically thick clouds, which they attributed to an excess

of cloud water in the model thick clouds. This approach is interesting because it allows a model to be evaluated over a large domain and a long period, but it does not distinguish between cloud regimes such as cloud altitude or phase, which could help better understand SWD errors. Köhler et al. (2017) had a different approach allowing for a distinction by cloud regime. They analysed the PV power forecast errors of NWP model COSMO-DE in Germany for 2013 and 2014 and highlighted that nearly one-third of the 100 days with the largest SWD errors were associated with fog and low stratus events. However, while this

study demonstrated the need for more reliable forecasts of low cloud cover, it was limited to 100 days and the cloud regimes were analyzed manually, which did not allow a systematic distinction of cloud situations. Other more systematic studies relied on highly instrumented sites with lidars and radars to detect the presence of clouds (Tuononen, 2019), or to automatically classify clouds based on their base and thickness (Ahlgrimm and Forbes, 2012). These approaches are useful for determining how cloud regimes and cloud physical parameters contribute to SWD errors in NWP models. More specifically, Tuononen

(2019) showed that low and mid-level clouds in the Integrated Forecast System (IFS) model in Helsinki are associated to a positive bias of SWD when the liquid water path (LWP) is low. They showed that an overestimation of SWD correlates with an underestimation of cloud fraction in the IFS model. Similarly, Ahlgrimm and Forbes (2012) estimated that at the ARM Southern Great Plains (SGP) site, in overcast low cloud conditions, the frequency of low LWP clouds was overestimated, and the frequency of high LWP clouds underestimated in the IFS model.

These highly instrumented sites also allow to evaluate other parameters in the absence of clouds. For instance Weverberg et al. (2018) analysed at the ARM SGP site the contributions of surface albedo, surface long wave emission, integrated water vapor, aerosols, and cloud properties in nine Global Circulation Models and found that cloud errors generally dominate SWD errors. Likewise, Morcrette (2002) found an underestimation of water vapor absorption, errors in humidity and in aerosol concentrations in the IFS model. Rieger et al. (2017) also reported large SWD errors during a Saharan dust outbreak over Germany in

the ICON NWP model.

However, most studies investigating the causes of SWD errors in NWP forecasts have used highly instrumented sites where detailed, high frequency observations are available, but this approach remains limited to a single site. In our study, we aim to develop a general method to evaluate SWD forecasts from NWP models and identify the situations that contribute most to errors. The method should allow to deal with a large domain with a sufficient number of reference observations, as in Nielsen

and Gleeson (2018). It should not be not limited to a single, probably unrepresentative site, as it is often the case when using a single supersite, and should allow to go further in identifying cloud errors. Our strategy is also to explore a wide variety of meteorological situations, which is possible by evaluating the model throughout the year, and not only during a short period where specific seasonal errors could dominate the overall behaviour. Finally, our method is intended to be general and systematic enough to be applied to any NWP model, relying only on SWD observations and making extensive use of cloud satellite

products.



In this paper, our evaluation methodology is applied to AROME at 1.3 km horizontal resolution. For this purpose, a full year of 1-day hourly forecasts is compared with *in situ* SWD measurements from the pyranometer network operated by Météo-France. The cloud products derived from the Satellite Application Facility for Nowcasting and Very Short Range Forecasting (SAF NWC) from geostationary satellites are used to classify cloud scenes at high temporal frequency and over the whole territory.

The paper is organized as follows. Section 2.1 presents the satellite products and SWD measurements, as well as the AROME forecasts used in this study. The evaluation methodology is then detailed in Section 2.2. Section 3 presents the results, first in terms of cloud occurrence, and then in terms of cloud situations. The results are then put into perspective in Section 4 where the limitations of the observations and potentials sources of error in AROME are discussed.

## 2    Data and methods

### 2.1    Observation and modeling data

#### 2.1.1    AROME forecasts

AROME (Applications de la Recherche à l'Opérationnel à Méso-Echelle) is a limited-area non hydrostatic model developed by the French weather service Météo-France (Seity et al., 2011; Brousseau et al., 2016). It covers a large part of Western Europe and has a horizontal resolution of 1.3 km. The number of model pixels is $1525 \times 1429$ and the number of vertical levels is 90.

The timestep is 50 s. The lateral boundary conditions are provided by ARPEGE (Action de Recherche Petite Echelle Grande Echelle), the French operational global NWP model.

The AROME model physical package is derived from the Meso-NH model (Lafore et al., 1998; Lac et al., 2018). The shallow convection is parametrized using the EDMF (Eddy Diffusivity Mass Flux) approach (Pergaud et al., 2009). The microphysical scheme is the one-moment mixed ICE3 scheme, completed with a subgrid condensation scheme presented in Riette and Lac

(2016). The turbulence parametrization considers a prognostic turbulent kinetic energy (TKE) equation from the Cuxart et al. (2000) scheme in a 1D mode, and is closed with the Bougeault and Lacarrere (1989) mixing length. The radiation parametrization comes from the IFS model, and comprises the six spectral bands shortwave radiation scheme of Fouquart and Bonnel (1980), and the Rapid Radiative Transfer Model for longwave radiation (Mlawer et al., 1997). AROME is coupled online with the SURFEX (externalized land and ocean surface platform) model, which describes the surface fluxes and the evolution of

four types of surfaces: natural, town, inland water and ocean (Masson et al., 2013).

In this study, 24-hour operational hourly forecasts starting at 00 UTC are evaluated, considering the average hourly SWD and average hourly total cloud fraction forecasts. We also use the average hourly cloud fraction in slices across the tropospheric column, located between the 1013 hPa - 785 hPa ($\sim$ 2 km), 785 hPa - 450 hPa ($\sim$ 6 km) and 450 hPa - top of model pressure levels. This study focuses on the year 2020 (corresponding to the AROME cycle 43t1). This year was chosen because it

corresponds to a recent and continuous period without major changes in the AROME operational model.





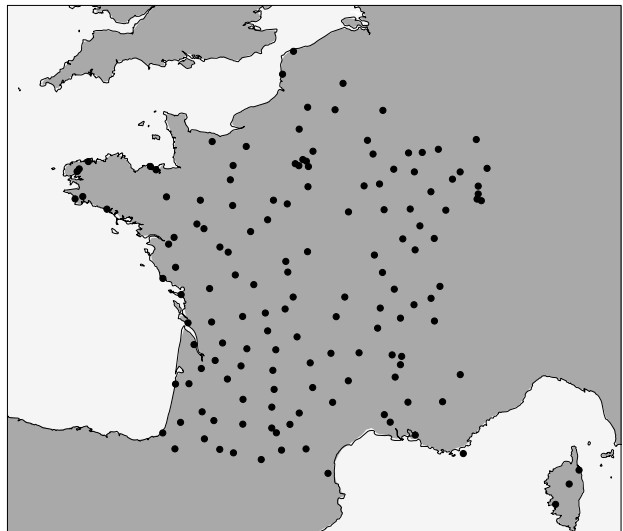

**Figure 1.** Localisation of the 168 used pyranometers from the network operated by Météo-France.

### 2.1.2 Observations of shortwave downward radiation

Within the operational observation network of Météo-France, 221 pyranometers in mainland France provide hourly means of SWD. Among them, only 168 were used during the whole year 2020 and considered of sufficient quality. Their locations are shown in Fig. 1. Practically, measurements taken under critical environmental conditions (e.g. with local masks, in mountainous regions above 1000 m) or with time series showing anomalies have been discarded to avoid introducing observational errors in the reference measurements.

### 2.1.3 Cloud satellite products

In order to identify the presence and type of clouds at high frequency and over the whole AROME domain, cloud satellite products developed by the NWC Eumetsat SAF are used (LeGleau, 2019; LeGléau and Kerdraon, 2019). Over France, the horizontal resolution is around 5 km and the temporal frequency is 15 minutes, so four values for one hour. The cloud mask product gives discrete values: 0 for no cloud and 1 for clouds. The cloud type is a classification among 15 different classes: 4 corresponding to cloud free scenes, 11 corresponding to clouds scenes (see Table 1).

## 2.2 Methods

### 2.2.1 Model – observations comparison of shortwave downward radiation

We compare hourly means of SWD from the model and from observations. To avoid grazing angles under which pyranometer measurements can be inaccurate, only the diurnal values corresponding to a cosine of the solar zenith angle (SZA) greater than





0.1 (equivalent to a SZA less than 84.3 °) are considered. To avoid double-penalty issues, usual for high resolution models like AROME that are unlikely to form clouds at exactly the right place and time (Amodei and Stein, 2009; Stein and Stoop, 2019), we do not necessary select the closest point, unlike most SWD evaluation studies, but instead extract it in the vicinity of the

observation point. In practice, a neighborhood strategy is set up that allows the selection of a model point with a forecast SWD value close to the observed value in a small square domain centered on the observation point. We sort the absolute model-observation errors in ascending order and select the 10$^{th}$ percentile value. This strategy avoids to select by chance a point very close to the observation, and ensures that the method is not too sensitive to the size of the neighborhood. Here, the size of the neighborhood is set to $5 \times 5$ model pixels which means that the point with the 2$^{nd}$ smallest difference (among 25 values) is

retained for each observation.

The error metrics commonly used to evaluate the SWD forecast errors are the mean error or bias (ME) and the root-mean-square error (RMSE). The ME describes systematic deviations and provides information on the sign of error. However, the ME is not sufficient to assess errors because of compensation effects. RMSE is more sensitive to large errors and thus particularly suited to the electricity market where large errors are much more critical than small errors (Perez et al., 2013; Betti et al., 2021).

RMSE can be divided into systematic errors (ME) and unsystematic errors or standard deviation of errors (SDE), as described in Widén et al. (2015); Antonanzas et al. (2016):

$$RMSE^2 = SDE^2 + ME^2 \qquad (1)$$

In practice, while it can be relatively easy to correct bias in a model, with a simple tuning or a statistical adjustment, reducing SDE is generally more challenging.

### 2.2.2 Cloud occurrence classification


To identify cloud presence in the observations, we use the satellite cloud mask. As AROME forecasts are only available as hourly means, comparable satellite products at hourly resolution are computed. Clear sky conditions are considered for a pyranometer location (i.e. the pixel satellite including the location of the pyranometer) at hour $H$ when the last four consecutive observations report no clouds. To identify clear sky conditions in AROME, the hourly mean total cloud fraction is computed as

the average over the neighborhood defined in the previous section, which size is comparable to the satellite pixel. A threshold value of 2 % is fixed: if the hourly mean total cloud fraction is larger than this value the hour is considered cloudy, and clear sky otherwise. Once cloudy skies are identified in both observations and model, we set up a contingency table, following Tuononen (2019): "hit" when clouds are present in the model and in the observation, "false alarm" when clouds are present in the model only, "miss" when clouds are present in the observation only, and "correct negative" when both observations and model agree

on clear sky conditions.

### 2.2.3 Cloud regimes and cloud types

In the model, to further discriminate between different cloudy situations, we use a modeled cloud regime classification similar to Weverberg et al. (2018). In AROME forecasts, the cloud fraction for each of the 3 distinct vertical slices is used. For each



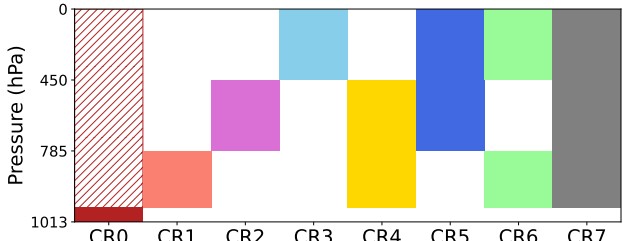

**Figure 2.** Modeled cloud regime classification.

region, a 2 % threshold is set to distinguish cloudy and clear sky conditions. Seven cloud regimes (CR 1 to 7) are defined from
the combination of these cloudy layers. When the liquid water content is larger than $10^{-5}$ kg kg$^{-1}$ at the first vertical level of
the model, the cloud regime is set to CR0, which corresponds to fog. These 8 cloud regimes are depicted in Fig. 2. An other
special cloud regime is defined for the situation when the total cloud fraction is larger than 2 %, so cloud are present in the
model, but the cloud fraction for each of the 3 vertical slices of the troposphere is lower than 2 %.

In the observations, to go further distinguish cloudy skies, we use a cloud type classification based on the cloud type satellite
product developed by SAF NWC and described in Section 2.1.3. As the values are instantaneous and discrete, we select the
values at H-30 minutes. The 11 cloud types of the products are merged into 6, allowing an easier comparison with model
outputs (Table 1).

| LC | Low clouds, which include very low and low clouds |
|---|---|
| MC | Mid-level clouds, which include mid-level clouds |
| SC | Semi-transparent clouds, which include high semi-transparent thin clouds, high semi-transparent moderately thick clouds, high semi-transparent thick clouds, high semi-transparent above low or medium clouds, high semi-transparent above snow/ice |
| HC | High clouds, which include high opaque clouds, very high opaque clouds |
| FC | Fractional clouds, which include fractional clouds |
| Others | Fill values or when the cloud type product indicates no clouds unlike the cloud mask product |

**Table 1.** Observed cloud type classification

Figure 3 shows the monthly relative frequency for each cloud regime (respectively type) and for the clear skies in the model
(respectively in the satellite product) over the pyranometers locations and when cos(SZA) > 0.1. In the model, the frequency
of clear skies is 22 % against 30 % in the satellite product, suggesting that the model predicts too many clouds or/and some
optically thin clouds are not detected by the NWC SAF product, which is a known caveat of passive sensors (Sun et al., 2011).
AROME predicts less low clouds (23 % of CR0+CR1+CR4) than satellite observations (29 % of LC). In addition, a seasonal
cycle is more visible in the observations than in the prediction with a relative frequency of observed fractional clouds higher





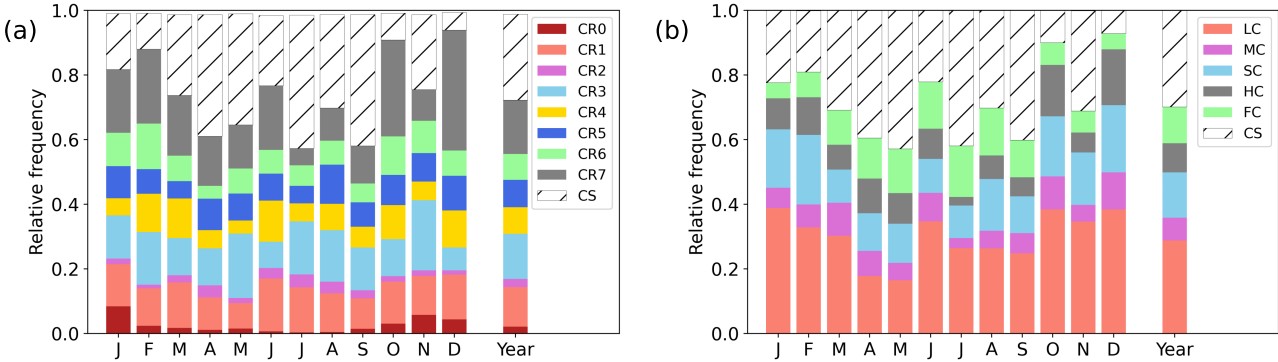

**Figure 3.** Monthly relative frequency for each (a) cloud regime in AROME and clear skies (CS), (b) cloud type in the satellite images and clear skies (CS), over 2020 for the pixels including the pyranometers.

in summer than in winter. We can also note that the model predicts too many thick clouds with a high cloud top (29 % of

CR5+CR7) compared to the observations (9 % of HC).

## 3   Statistical analysis

The method presented in the previous section is now applied to evaluate the AROME SWD forecasts, starting with an overall evaluation before going into detail.

### 3.1   Overall evaluation: clear sky index histograms

To begin with, histograms of clear sky index (CSI), the ratio of SWD (in the model and in the observation) divided by the theoretical SWD under clear sky conditions from the model, are used to point differences in SWD between AROME and observations (Fig. 4), similarly to what Nielsen and Gleeson (2018) did to evaluate the NWP model HARMONIE-AROME against the Danish pyranometers network. The CSI roughly quantifies overall cloud transmittance, bearing the signature of both cloud cover and cloud optical thickness. Only values for which the SZA is less than 70 ° are considered to ensure that the

CSI distribution is not affected by measurement errors and model limitations at grazing angles.

In the observations the CSI can greatly exceed 1, which can be due to an underestimation of the SWD during clear skies conditions in the model (Nielsen and Gleeson, 2018) or to cloud enhancement effects (Gueymard, 2017). Such effects cannot be simulated with standard plane parallel radiative codes (Wissmeier et al., 2013). The few CSI values in the model exceeding 1 are unrealistic and due to the fact that the theoretical SWD under clear sky conditions from the model isslightly delayed

compared to the SWD. The CSI in the model is more frequently between 0.8 and 1 compared to observations (which may partly be due to the lack of values exceeding 1 in the model), or less than 0.1, and in contrast less frequently between 0.1 and




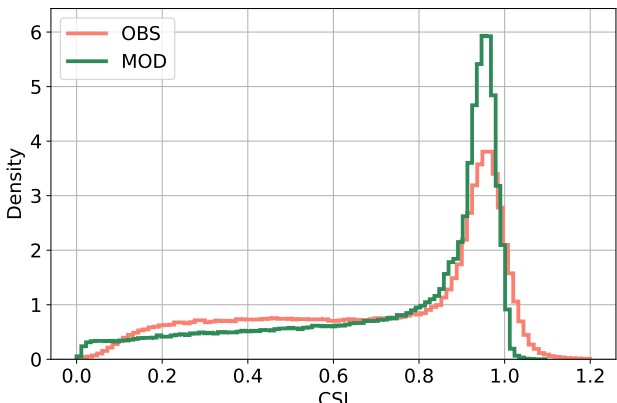

**Figure 4.** Clear sky index distribution in the model and in the observations when SZA > 70° over 2020.

0.75. It suggests that optically thick clouds are too thick, which agrees with the excess of CR5 and CR7 pointed in Fig. 3. Interestingly the CSI distributions are similar to those reported by Nielsen and Gleeson (2018) with HARMONIE-AROME, which suggests that NWP model errors are to some extent systematic over a wide range of situations. The opposite behaviours
in different ranges of CSI show that errors in AROME are not limited to a systematic bias which could be easily corrected with a rough tuning. Instead, the behaviours seem to depend on cloud situation, which will be further investigated in the following.

### 3.2 Attributing model errors to cloud occurrence

To go beyond Fig. 4, we now investigate how the different cases of cloud presence defined in Section 2.2.2 contribute to the overall errors. To this end we compute the relative frequency, SWD bias and SDE for each of the four occurrence cases. To
identify the situations that contribute most to errors, another metric is used: the contribution to the total bias, which is the frequency weighted bias. The sum of these contributions equals the total bias. If a situation is associated with a high bias but rarely occurs, it barely contributes to the total bias. On the contrary, a situation associated with a low bias can significantly contribute to the total bias if it frequently occurs. Figure 5a shows the monthly bias, frequency and contribution to the total bias.
As expected, the bias is negative for the false alarm cases (ranging from -5 to -25 W m$^{-2}$), and positive for the miss cases (with monthly biases up to 80 W m$^{-2}$). The false alarm cases are almost three times more frequent than miss cases (11% of false alarm cases and 4% of miss cases during the year 2020), which is consistent with the higher clear skies frequency in the observations pointed in Section 2.2.3 and suggests that AROME predicts too many clouds or that some clouds are not detected by the satellite. With a cloud fraction threshold value of 10 %, the gap between the two frequencies is reduced but we still
have more false alarm cases (10 % of false alarm cases and 6 % of miss cases). The bias is stronger for the miss cases than for the false alarm cases. This suggests that clouds missed by AROME have more impact on the SWD than those simulated in false alarm cases, or that some false alarm cases are actually hit cases with undetected clouds. When both the model and





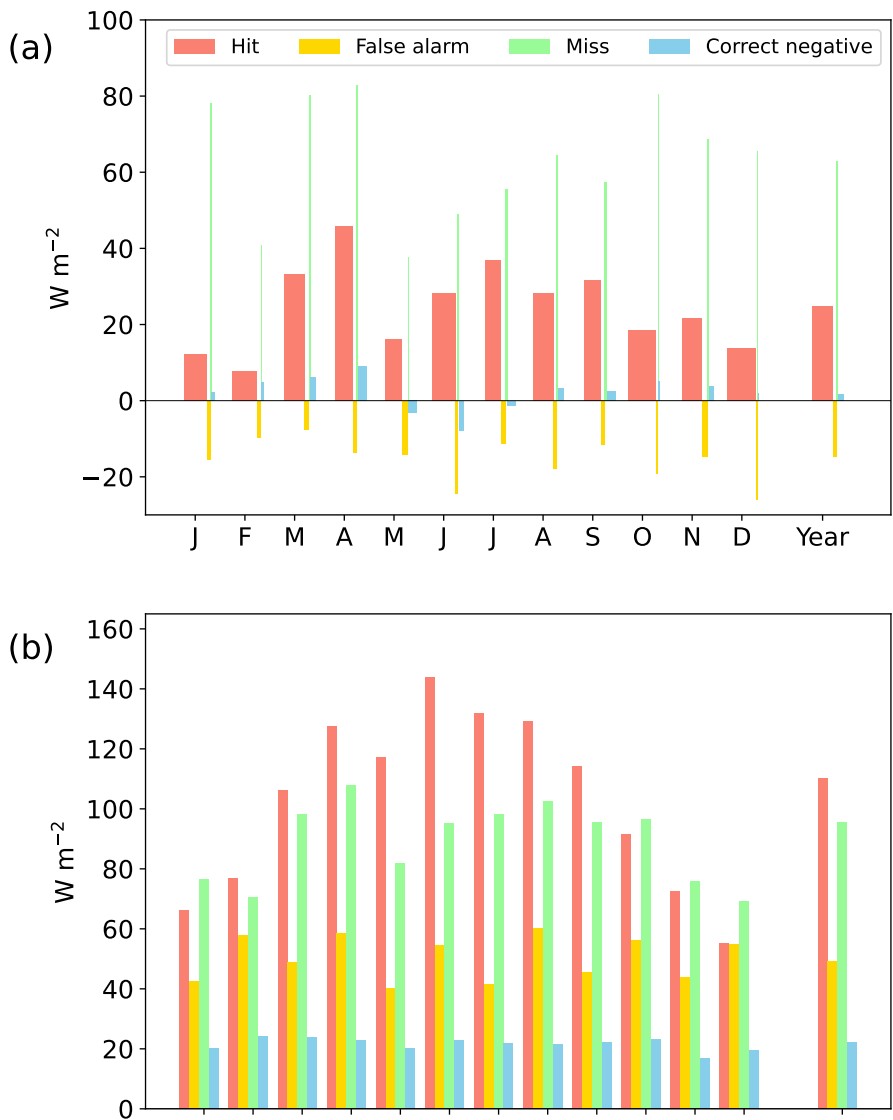

**Figure 5.** (a) Monthly mean SWD bias (bar height, in W m$^{-2}$) and relative frequency (bar width) for each category during the year 2020: red for the hit cases, yellow for the false alarm cases, green for the miss cases and blue for the correct negative cases. The total contribution is the bar surface. (b) Monthly SWD standard deviation of errors (SDE) for each category during the year 2020.

observations predict clear skies, the bias is much smaller, ranging from -10 to 10 W m$^{-2}$, and is on average slightly positive during the year. The hit cases represent 66 % of the situations and are associated with significant biases up to 50 W m$^{-2}$. Their contribution to the total bias is the most important (16 W m$^{-2}$, for a total bias of 18 W m$^{-2}$). Interestingly, the bias associated with hit cases is invariably positive throughout the year, showing a tendency of AROME to underestimate the impact of clouds





on the SWD. Figure 5b shows that the SDE is also the strongest for the hit cases, with annual values of 110 W m$^{-2}$, compared to 50 W m$^{-2}$ for the false alarm cases, 95 W m$^{-2}$ for the miss cases and 22 W m$^{-2}$ for the correct negative cases. Note that several tests were performed by changing the neighborhood size and/or the cloud detection threshold, which did not change
theses results. Hence we consider these results are robust and independent of the mentioned thresholds.

To summarize, errors mostly occur when clouds are present in the model and in the observations, which is consistent with Weverberg et al. (2018) and Tuononen (2019).

### 3.3   Attributing model errors to AROME cloud regimes and satellite cloud types

In what follows, we provide a somewhat more physical overview of correct negative cases, false alarm cases and miss cases
results. Then we focus more extensively on the hit cases that contribute most to SWD errors. To this end, we distinguish between situations in terms of cloud regimes in AROME or in terms of the satellite's cloud types, so that bias and SDE can be attributed to these specific cloud situations.

#### 3.3.1   Correct negative cases

Various factors can explain SWD errors for the correct negative cases. For example, Morcrette (2002) has found a persistent
positive bias in clear sky that they attributed to an underestimation of gaseous absorption in the IFS water vapor spectroscopy, which is still the one used in AROME. Another factor which can contribute to SWD errors in clear sky condition is relative to aerosols. In AROME, the aerosols are prescribed by a monthly climatology (Tegen et al., 1997), meaning that only the average seasonal cycle of aerosols is captured, but the individual aerosol events are not accounted for. Such events, for instance dust outbreaks, can result in local and punctual SWD attenuation of 40-50% as pointed by Kosmopoulos et al. (2017), so not taking
them into account can lead to significant SWD errors (Rieger et al., 2017).

To estimate the extent to which such a misrepresentation of aerosols contributes to the errors of the correct negative cases, we examine the correlation between SWD errors and aerosol optical depth (AOD) errors, the latter being defined as the difference between the climatological AOD from AROME and the AOD estimated from Copernicus Atmosphere Monitoring Service
(CAMS), a chemistry-transport model that provides near-real time AOD forecasts and fully captures aerosol-related events (Benedetti et al., 2009; Morcrette et al., 2009), interpolated to the AROME grid. Figure 6 shows the natural logarithm of the transmittance (T) (SWD/SWD$_{TOA}$ with SWD$_{TOA}$ the SWD at the top of the atmosphere) errors as a function of AOD errors divided by cos(SZA) in the correct negative cases, as well as the linear regression line. Again, only values for which the SZA is less than 70 $^\circ$ are considered in order to avoid too low values of SWD$_{TOA}$.


As expected, a negative slope is obtained during the year 2020 with a value of -0.07, but also for each individual month (not shown). The correlation coefficient is low, showing a large variability in the data, with a value of -0.19 during the year, and lower values in winter and higher values in summer. However, throughout the year the p-value is very low, indicating a statistically significant relationship between SWD errors and AOD errors. The low correlation coefficient could be explained by the





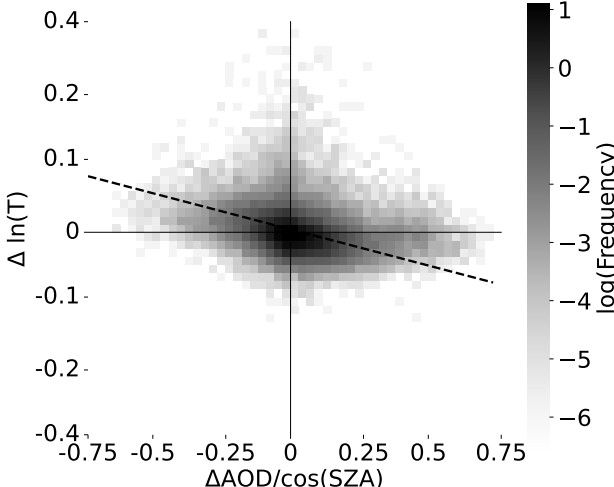

**Figure 6.** ln(Transmittance ($SWD/SWD_{TOA}$)) errors as a function of AOD errors with the fit line during clear skies in year 2020 when the SZA is less than 70 °

different factors mentioned above that explain SWD errors, but mainly by clouds which are no detected by cloud products but alter the SWD (see section 4.1.2). Using passive measurements for cloud detection limits our study of aerosol errors, because cloud detection errors influence clear sky SWD errors at the first order. A better cloud detection is required to identify real clear skies in the reality, as discussed in section 4.1.2.

We do not extend further on these clear sky cases which have been treated elsewhere, notably Rieger et al. (2017) who show an overall improvement of the PV-power forecast during a Saharan dust event over Germany in the ICON model when extended with modules accounting for trace gases and aerosols and related feedback processes. Moreover, the Tegen climatology used in AROME is outdated and a more recent climatology as the CAMS climatology could reduce SWD errors, as implemented in the IFS (Bozzo et al., 2020).

### 3.3.2 False alarm cases

False alarm cases correspond to clouds predicted by AROME but not observed by the satellite. We study here in more detail the type of clouds present in AROME in such cases.

Figure 7a shows that the relative frequency for geometrically thin clouds in AROME, designating cloud regimes with clouds in only one vertical slice of the troposphere, is higher when no clouds are detected than in the AROME total cloud climatology (for instance 50 % vs 18 % for CR3, see Fig. 3a). This suggests that some thin clouds (mostly high clouds) formed by AROME are not physically realistic, or are not detected by the satellite. Regarding the biases, they are mostly negative for most of the cloud regimes, except for CR3 for a few months, CR3 having the lowest absolute bias throughout the year. This also shows that some geometrically thin high clouds may not be detected. The major contribution to the total bias in false alarm cases is



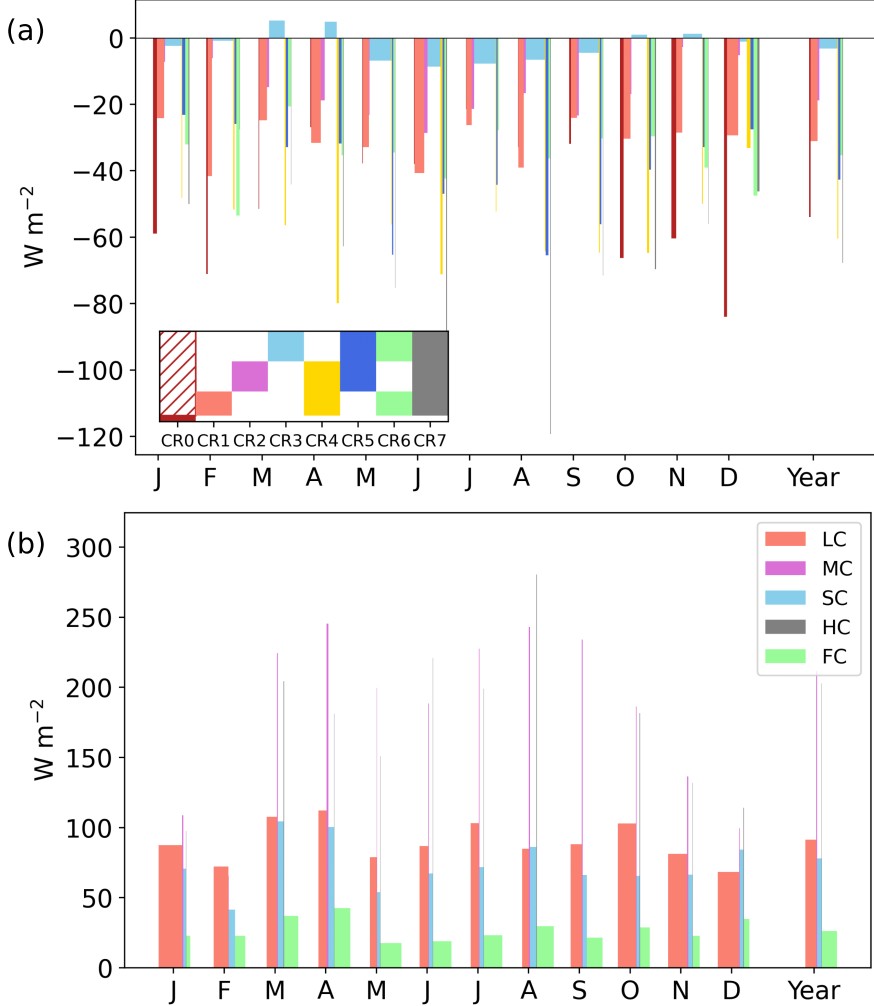

**Figure 7.** Monthly relative bias (bar height) and frequency (bar width) for (a) each cloud regime for the false alarm cases, (b) each cloud type for the miss cases. The vignette of the cloud regimes is inserted to facilitate the reading.

for geometrically thin low clouds (CR1) which have a lower frequency than CR3 but a much larger bias throughout the year.
Given the SWD errors associated with CR1, it is likely that some are erroneously predicted by the model. In contrast, almost
no geometrically thick clouds (CR4, CR5 and CR7) are simulated when no clouds are observed (5 % for CR4, 6% for CR5 and
2.7 % for CR7), while they are predominant in the total climatology (13 % for CR4, 11% for CR5 and 28 % for CR7). This is
consistent with the fact that simulating thick clouds in the absence of clouds is unlikely and that thick clouds are less likely to
be undetected.





Around 4 % of the false alarm cases are fog, compared to about 3 % of simulated clouds in total. In winter, there can be up
to three times more fog in the false alarm cases than in the total AROME cloud climatology, suggesting that during the day
in winter, there is a lot of simulated fog while no cloud is detected. As fogs are largely present in the morning (7.6 % in the
morning and 0.5 % in the afternoon during the year), this may be due to the delay in dissipation of AROME fog highlighted
by (Antoine et al.). In winter, the highest negative bias is for fog (minimum of -83 W m$^{-2}$) with a relatively high frequency,
showing that the contribution of fog to the bias is particularly important in winter and that improving AROME fog forecasts is
an important issue for SWD errors.

### 3.3.3    Miss cases

As for the false alarm cases, the objective is to identify which cloud types have been missed by AROME. Figure 7b shows that
the seasonal cycle is more pronounced than for the false alarm cases, not visible in the observations in Fig. 3b. In winter, the
relative frequency of low clouds (LC) in the miss cases is much higher (maximum of 74%) than for the observed climatology
as a whole (41 %, see Fig. 3b), while the opposite is true in summer with a minimum of 21 %. Figure 7b also shows that, as
expected, the bias for all cloud types is positive and that the main contribution to the bias for the year 2020 comes from LC,
especially in winter and autumn.

    The annual relative frequency of fractional clouds (FC) in miss cases is the highest (47 %), especially in summer when they
are more frequent with a maximum of 67 %. Regardless of the season, the relative frequency of FC in miss cases is higher than
in the total observed climatology (maximum of 27 %). In summary, it is mainly LC that are missed by AROME in winter and
mainly FC in summer.

    Few high clouds (HC) are missed, with an annual relative frequency of 1 % (13 % in the total observed climatology), unlike
semi-transparent clouds (SC) which are often missed with an annual relative frequency of 14 % (20 % of the total observed
climatology). The highest biases come from mid-level clouds (MC) (211 W m$^{-2}$) and HC (203 W m$^{-2}$) but due to their very
low occurrence, their contribution to the bias in miss cases is very small, in contrast to SC.

### 3.3.4    Hit cases: in all situations

We now focus on the hit cases, which are the most frequent and contribute the most to the overall errors. We investigate how
in these cases the errors depend on the AROME cloud regimes and satellite cloud types. Figure 8a shows the monthly bias and
relative frequency as well as the contribution to the total bias for the hit cases over 2020. It appears that AROME overestimates
the SWD for almost all CRs, except for CR0 in all the months, and CR4 and CR7 in some months. The largest positive bias is
for CR3, with values up to 93 W m$^{-2}$. Figure 8b shows the same behaviour for almost all cloud types, except for FC. Such a
positive bias could be explained both by a systematic underestimation of the cloud fraction and/or by an underestimation of the
cloud optical thickness. Since we do not have an estimate of the cloud fraction, it is difficult to conclude. Indeed, the evaluation
of the cloud fraction at such a high spatial and temporal resolutions over a large domain remains a challenge.

    To go further, Figure 9 shows the bias for each cloud regime and four ranges of forecasted cloud fraction. At low cloud
fraction (i.e. less than 10 %), the bias, in addition to being the highest for almost all cloud regimes, is positive for all cloud



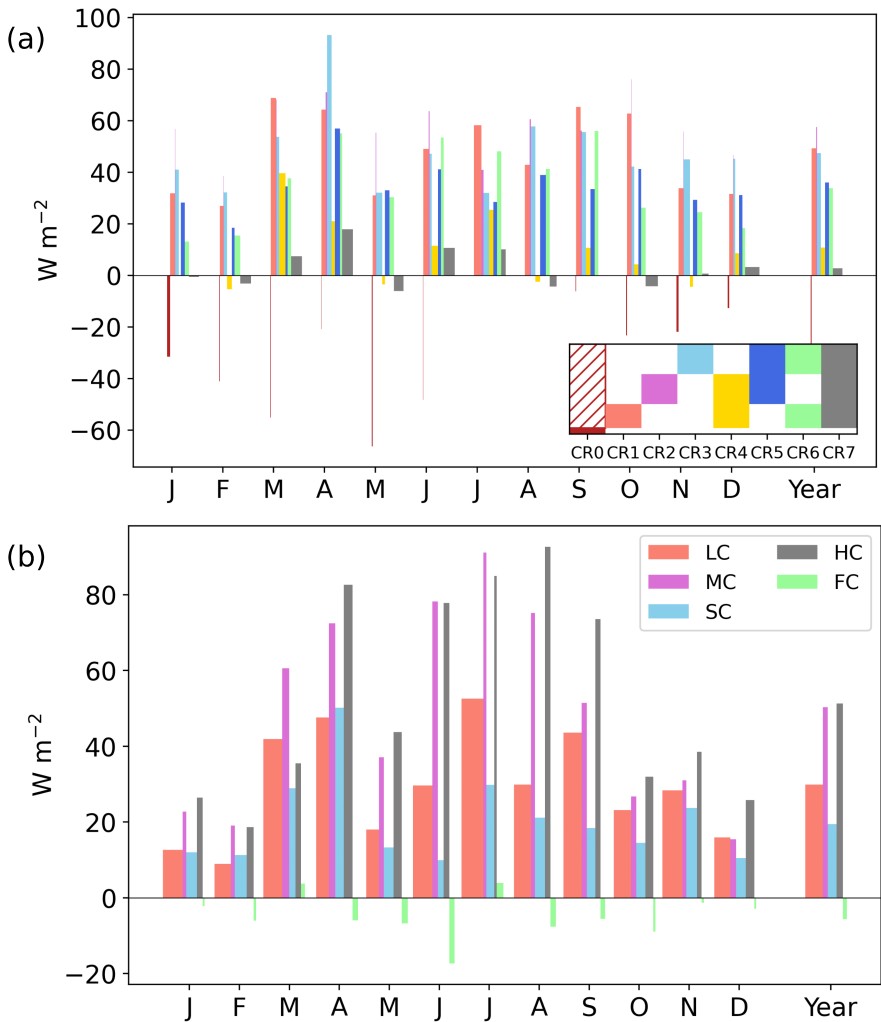

**Figure 8.** Monthly mean SWD bias (bar height, in W m$^{-2}$) and relative frequency (bar width) for each (a) cloud regime and (b) cloud type, during the year 2020 for the hit cases. The total contribution is the bar surface.

regimes. This is not surprising since when the cloud fraction is low in the model, if the cloud fraction is not correct, it is most likely to be underestimated, resulting in a positive bias. For a high cloud fraction (more than 95 %), the bias remains

positive for CR2, CR3, CR5 and CR6, despite the fact that in this case, if the cloud fraction is not correct, it is most likely to be overestimated, which would lead to a negative bias. This suggests that for these cloud regimes, errors are not only governed by cloud fraction errors.

     This preliminary analysis highlights errors that can be attributed to cloud fraction errors. Since we do not have access to a reliable cloud fraction observation for the AROME resolution and domain, we do not pursue the evaluation under all cloudy





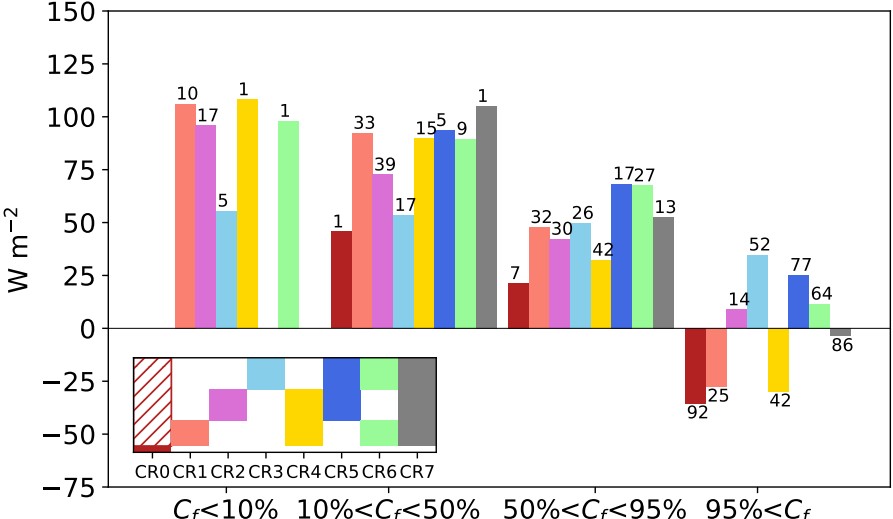

**Figure 9.** Annual mean SWD bias for different AROME cloud fractions and for each cloud type during the year 2020. The number above each bar represents the frequency (in %) by cloud fraction range for each cloud regime.

conditions. Instead, from now on we focus on situations where the AROME cloud fraction is close to unity, which means that positive biases cannot at least be attributed to an underestimation of the cloud fraction.

### 3.3.5 Hits cases: in overcast situations

As explained, we now focus on overcast situations. In practice, we consider the hit cases for which the AROME cloud fraction is larger than 95 %, which corresponds to 61 % of the hit cases, and 40 % of all the cases. The frequency of overcast situations for each cloud regime is shown in Fig. 9 (black numbers in %). In practice, these proportions are largely the same when the overcast threshold is set between 95 % and 99 %. The first effect of this overcast sub-selection is to change the annual bias, leading to a nearly-zero bias (1.1 compared to 24 W m$^{-2}$ before). This suggests that part of the overall positive bias may be due to errors in cloud fractions or too low optical thickness under partial cloudiness. On the contrary, the SDE is barely reduced (101 vs 110 W m$^{-2}$), showing that the overcast cases still deserve attention, and that the nearly-zero bias results from error compensations. In this section, we investigate how the errors vary with the AROME cloud regimes and satellite cloud types.

We start by distinguishing these overcast hit cases in terms of modeled cloud regime, or observed cloud type in the satellite classification. Figure 10a shows the 2020 monthly bias, relative frequency, and contribution to total bias.

When the situations are sorted according to the AROME cloud regime, the high clouds (CR3 and CR5) are systematically associated with a positive bias throughout the year, with values up to 100 W m$^{-2}$. Since these two cloud regimes correspond to 21 % of the cases, this results in a significant positive contribution to the bias. On the contrary, the bias for the low clouds (CR1 and CR4) is negative with a more pronounced seasonal cycle, characterized by a stronger bias in summer up to -80 W m$^{-2}$. Fog cases are associated with negative biases with an annual bias of -25 W m$^{-2}$ and a higher frequency in January. The annual



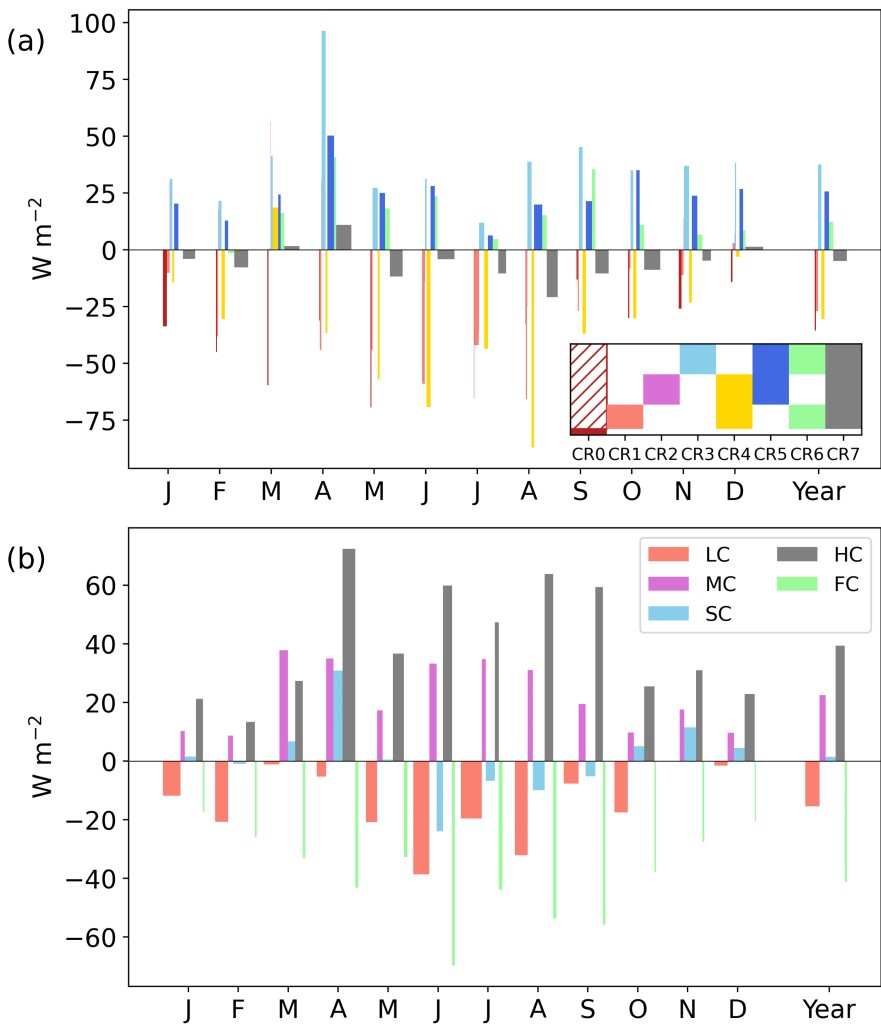

**Figure 10.** Monthly mean SWD bias (bar height, in W m$^{-2}$) and relative frequency (bar width) for (a) all modeled cloud regimes and (b) all observed cloud types, in overcast situations, during the year 2020. The total contribution is the bar surface.

bias is consistent with the already mentioned delay in fog dissipation and the too thick optical depth, related to the excess of water content and droplet concentration in the fogs simulated by AROME, as pointed out by (Antoine et al.).

When satellite cloud types are used (Fig. 10b), we again obtain a positive bias for HC, with values up to 70 W m$^{-2}$, a bias of variable sign for SC, and a persistent negative bias for LC with values up to -40 W m$^{-2}$. The bias is always negative for FC. This is consistent with the fact that this cloud type probably corresponds to situations with a cloud fraction lower than 100 %, meaning that the cloud fraction, if not correct, is overestimated in AROME. Note that satellites with passive instruments cannot





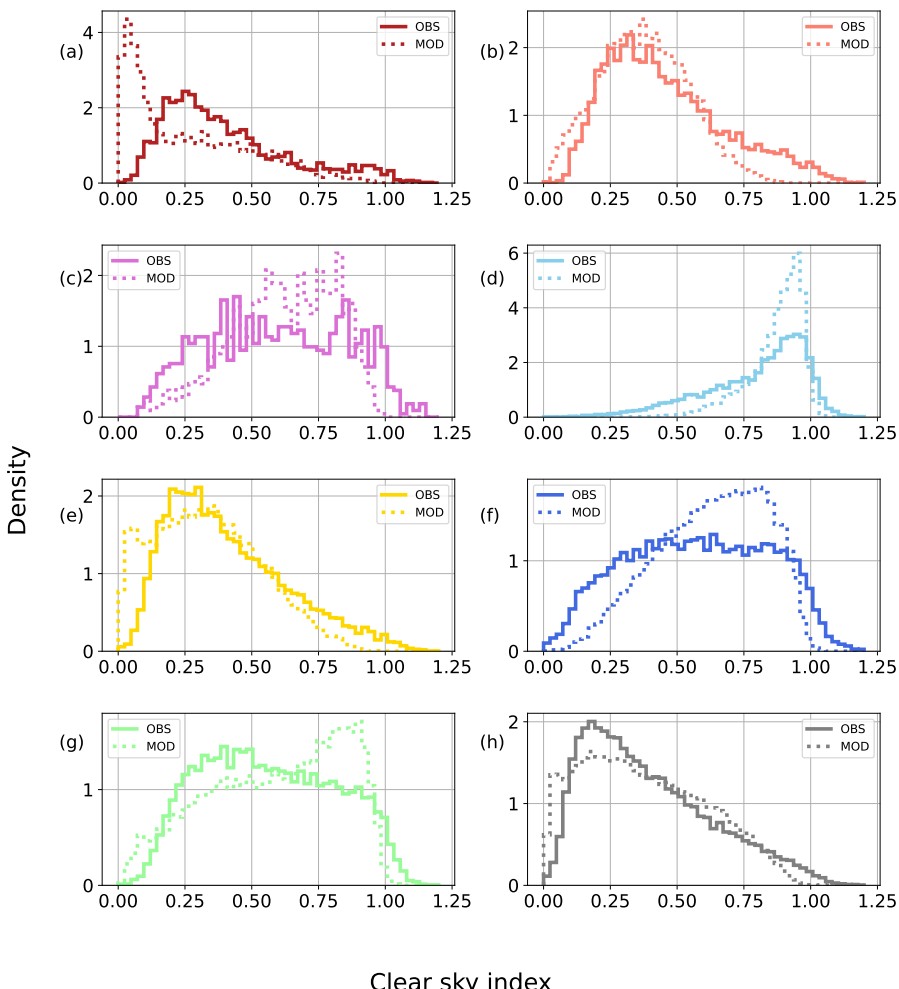

**Figure 11.** Distribution of Clear Sky Index for (a) CR0, (b) CR1, (c) CR2, (d) CR3, (e) CR4, (f) CR5, (g) CR6, (h) CR7 in AROME and in the observations when the SZA is less than 70 °.

detect low clouds when a high cloud layer is present, and therefore the modeled cloud regimes and observed cloud types are
not directly comparable.

The results are essentially the same when the cloud fraction threshold for defining the cloud regimes, applied independently over each vertical slice of the troposphere as explained in section 2.2.3, is set to 1 %, 2 % or 10 %.

Despite this, these results, largely consistent between the AROME and satellite classifications, highlight that although on average the SWD bias is small under overcast conditions, it results from the compensation of large and systematic errors,
positive for high clouds and negative for low clouds. This also explains why the SDE remains large compared to the bias.

To further investigate the error distribution for each cloud type, it is useful to examine the CSI for the AROME cloud regimes in overcast situations. Figure 11 shows the CSI distribution for all cloud regimes in the model and in the observations. For CR0,





CR1, CR4, CR6, and CR7, i.e. for the cloud regimes which have a non-zero cloud fraction in the lower atmosphere, the model has too many low CSI values below 0.1 compared to the observation. For CR1, and CR4, the model has not enough values

greater than 0.6. These two findings suggest that the optically thick low clouds are too thick and that the model does not have enough optically thin low clouds in overcast situations. This explains the overall negative bias for these cloud regimes. For CR3, CR5, and CR6, the model has too many high values (between 0.8 and 1 for CR3, between 0.5 and 0.9 for CR5 and between 0.6 and 1 for CR6), and for CR3, and CR5 not enough low values. This suggests that high clouds are overall optically too thin in AROME. Even though the model does not have CSI values greater than 1, while the observations do, the bias is

positive for these cloud regimes, as seen in Fig. 10a. This positive bias is not due to a potentially too small cloud fraction in the model, as only overcast situations in the model are selected.

Figure 12 shows the distribution of SWD errors for all AROME cloud regimes. For each CR The SDE is high compared to the mean flux, in particular for CR2, CR4, CR5 and CR6. The distributions show the contributions of both positive and negative errors, indicating that multiple sources of error are involved. The analysis are the same for the distribution of SWD errors for

observed cloud type in the satellite classification (not shown). More detailed observations are needed to further analyze these errors and their potential sources.

In conclusion, the errors in overcast conditions depend on the cloud regimes. Low clouds seem on average optically too thick, although the negative bias may be due to an overestimation of the cloud fraction in overcast situations in the model. On the contrary, simulated high clouds are often optically too thin. Although it would be interesting to evaluate the model over a

longer period, the fact that our results are consistent for each month of 2020 suggest that they are not specific to this particular year.

## 4   Discussion

In this section, we first address the cloud fraction issue and provide a critical analysis of the satellite cloud mask and cloud type. We then investigate potential sources of error in AROME that could explain the observed SWD errors.

### 4.1   Limitation of the observations

#### 4.1.1   Cloud fraction

In this study we did not evaluate cloud fraction, so we could not attribute SWD errors to potential cloud fraction errors. Indeed, evaluating the cloud fraction at AROME resolution over a large domain is challenging because spatially and temporally resolved cloud fraction observations at such spatial resolution are not common.

The cloud fraction of AROME at 2.5 km horizontal resolution has already been evaluated at larger time scale between 2000 and 2018 with the 0.05° COMET and 0.25° CLARA satellite products by Lucas-Picher et al. (2022). They show that in summer AROME underestimates the cloud fraction over land, resulting in an overestimation of SWD, while it overestimates the cloud fraction in winter and spring. SWD is also overestimated in spring and fall, although less than in summer. Like (Lucas-Picher



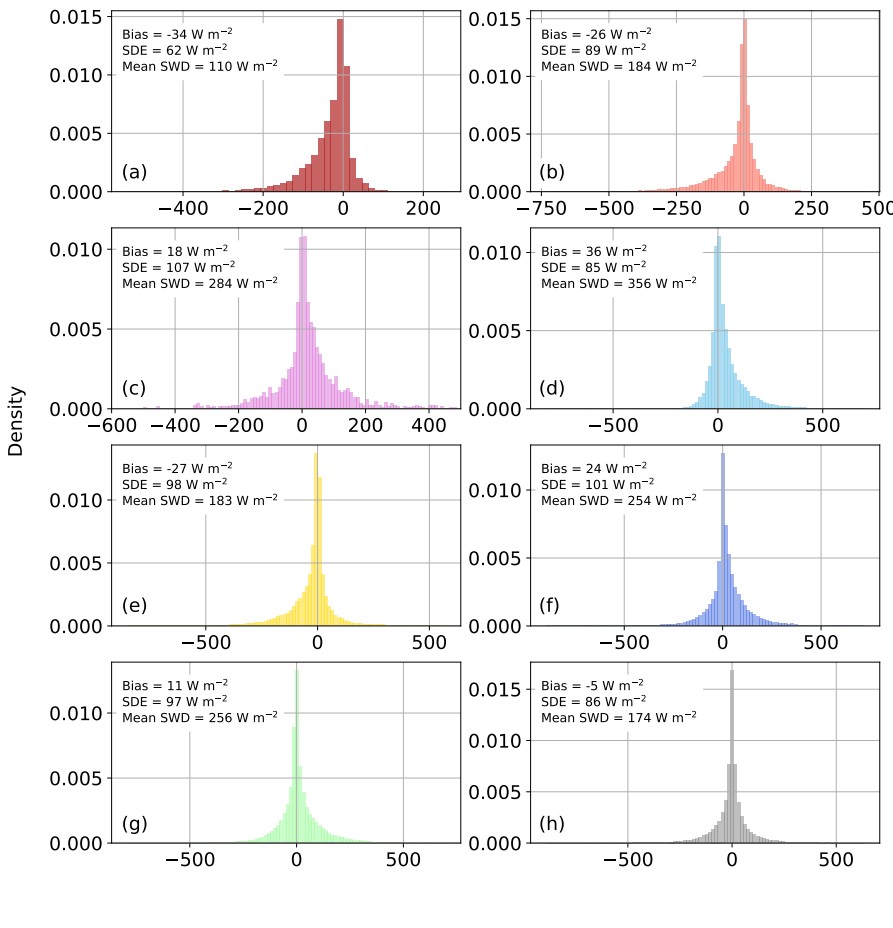

**Figure 12.** Distribution of SWD errors (bias and SDE indicated) for (a) CR0, (b) CR1, (c) CR2, (d) CR3, (e) CR4, (f) CR5, (g) CR6, (h) CR7 cloud regimes in overcast situations.

et al., 2022), we found a SWD positive bias for every month of the year (Fig. 5a) even in winter and spring when the cloud
fraction is possibly overestimated. This suggests that other sources of error than the cloud fraction are responsible for the
positive SWD bias, and that the underestimation of the cloud fraction in summer further accentuates the bias.

Satellite product at finer resolution should be used to analyze cloud fraction errors in AROME at 1.3km resolution over the
whole year 2020, for instance the daily MODIS product (Ackerman et al., 2008). An alternative to satellite observations would
be to use ground observations from instrumented sites, allowing evaluation at a finer scale.Cloud fraction can be estimated
from shortwave broadband measurements (Long et al., 2006), all-sky imaging systems (Pfister et al., 2003), or remote sensing
instruments such as radar, lidar and microwave radiometer (e.g. Illingworth et al., 2007).





However, the cloud fractions derived from different observations generally differ (Long et al., 2006; Wagner and Kleiss, 2016), and more critically, the cloud fraction is somehow loosely defined, so that the observed cloud fraction does not necessarily match the model definition (Brooks et al., 2005). Above all, assessing the performance solely on a few sites would not 405 be consistent with the evaluation strategy adopted in this study.

### 4.1.2 Cloud detection by satellite

In this paper, several hints suggest that some clouds are not detected by the satellite product, such as the asymmetry between the frequency of clear skies in the model and in the observations (Section 2.2.3), or between the frequency of false alarm and miss cases (Section 3.2). This questions the reliability of the satellite cloud mask used. In this Section, we explore other 410 indications of the cloud mask deficiencies and their potential impact on our results.

Figure 13a shows the annual mean transmittance and Fig. 13b the standard deviation (SD) of the atmospheric transmittance for all cases of cloud presence. Since clouds are the primary drivers of SWD and its variability, we assume that a lower mean SWD and a higher SD indicate the presence of clouds. Observation values, with and without detected clouds, are shown on the left side and model values on the right side. Comparable values are shown side by side.

The first pair of values shows that for the observed values when clouds are present only in the model (i.e. false alarms), the transmittance is slightly lower (-5 %) and the SWD higher (15 %) than for the correct negative cases, while the values in the observations should be about the same in these cases and should correspond to the model clear-sky. This suggests that clouds may be present but not detected, in particular in the false alarm cases, when clouds are simulated. Actually the fact that for the correct negative cases the observed SD is higher (29 %) than the model SD also suggests the presence of undetected clouds, 420 which can explain the large variability noticed in Fig. 6.

The likely non-detection of clouds may have an impact on our results. For the correct negative cases, this could explain a SWD positive bias in AROME, with clouds present in reality but not detected, and an overestimation of the frequency of correct negative cases. In the same way, it may lead to an underestimation of the hit cases frequency, in addition to not taking into account some optically thin real clouds in the error calculations. However, the impact on the hit cases is probably limited 425 due to their high relative frequency compared to the probably low relative occurrence of non-detection. In the future, a more reliable cloud detection could be based on ground observations from remote sensing instruments or high frequency global and diffuse SWD measurements Long and Ackerman (2000).

The second pair of values corresponds to the values of observations when clouds are detected, with either non-simulated clouds (miss cases) or simulated clouds (hit cases). When no clouds are simulated, the annual mean SWD (respectively SD) 430 for miss cases is 47 % higher (respectively 15 % lower) than for hit cases, suggesting that the clouds missed by the model are those that have on average a small impact on the SWD.

The fourth pair of values, corresponding to the model values when the clouds are simulated, shows that the transmittance (respectively SD) for the false alarm cases is 65 % higher (respectively 40 % lower) than for the hit cases. In addition, the model transmittance for the false alarm cases and for the correct negative cases are relatively close. This suggests that undetected 435 simulated clouds have on average less impact on the SWD than actually observed clouds.





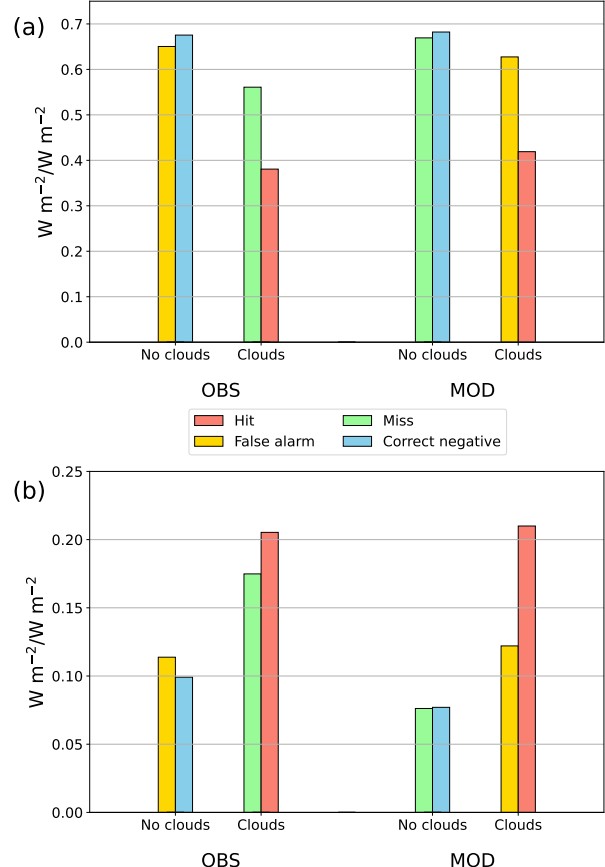

**Figure 13.** (a) Mean annual transmittance for each case in the model and in the observations during the year 2020 and (b) standard deviation of transmittance for each case in the model and in the observations during the year 2020. Transmittance is the SWD normalized by the SWD at the top of the atmosphere.

These results, in addition to pointing that clouds may be present but not detected by the satellite, also suggest that clouds missed by the model are generally clouds with a small impact on SWD, and in the same way, that simulated undetected clouds have a limited impact on the SWD. This conclusion is supported by Fig. 7 which shows a low frequency of geometrically thick clouds (CR4, CR5 and CR7) in the false alarm cases and a low frequency of HC in the miss cases. It should be noted that there

440 may also be cases of clouds detected but not present in reality (LeGléau and Kerdraon, 2019), but that the impact is probably minimal in our study.

## 4.2 Cross-comparison between the cloud regimes and the cloud types

To assess the consistency between simulated CRs and observed CTs and analyze the matches between the model and the observations, a cross-comparison is shown in Fig. 14. It displays the distribution of cloud types for each cloud regime, and





445 conversely, for the hit cases in overcast situations. Recall that the parallel between cloud regime and cloud type is not direct since satellite products do not see what is under an optically thick cloud.

For low clouds, Fig. 14a shows that for CR0, CR4 and mainly for CR1, LC is most often observed. On the contrary, Fig. 14b shows that when LC is observed, CR7 and CR6 are most often simulated, more than CR0, CR1 and CR4. CR2, CR3 and CR5 remain very rare in this case. The relative high frequencies of CR6 when LC are observed (18.8 %) and of LC when CR6 450 is simulated (50.9 %) can be explained by the non-detection of high clouds. Note that in Section 2.2.3, we have shown that CR7 are far too frequent in the model compared to observation (HC) and that it is therefore not surprising to find an excessive occurrence of CR7 in some cloud types. Even so, this suggests that for low clouds, the cloud type is in reasonable agreement with the model, and thus the negative bias of SWD found in section 3.3 is actually related to poorly simulated low clouds. The study of fog remains limited by satellite because the satellite product includes fog in low clouds (LeGleau, 2019), as fog and 455 low stratus are difficult to distinguish (Bendix et al., 2005), .

For optically thin high clouds, Fig. 14a shows that for CR3, SC (48.4 %) and FC (26.1 %) are mostly observed, which means a good match for optically thin high clouds since the distinction between SC and FC is questionable. Indeed, very thin cirrus are often classified as fractional clouds (LeGleau, 2019). Conversely, in Fig. 14b for SC, mostly of CR7 (36.5 %), CR5 (22.0 %) and CR3 (19.9 %) are simulated, so the relative frequency of CR7, as for LC, seems too high. For FC, CR3 is mostly 460 simulated (32.5 %).

For optically thick high clouds that most likely correspond to CR5, CR7 and HC, Fig. 14a shows that for CR5, SC and HC are mainly observed (63,8 % when added together), much more than LC, MC and FC. For HC, CR5 (20.3 %) and CR7 (68.0 %) are mainly simulated while few CR0, CR1 and CR4 are simulated. This means that we also have a relatively good match for high opaque clouds (HC) and geometrically thick high clouds in the model (CR5, CR7). Thus, for high clouds, the 465 model cloud regimes are in reasonable agreement with the observed cloud type, even though CR7 is too frequent in the model. This suggests that the positive SWD bias found in section 3.3 is indeed related to high clouds.

In summary, there is generally a good correlation between the modeled cloud regime and the observed cloud type, although the match is not perfect. When the model predicts low clouds, mostly low clouds are observed, and when the model predicts high clouds, mostly semi-transparent and opaque clouds are observed. However, CR7 seems too frequent, with in particular too 470 many occurrences when LC are observed. Note, however, that some of the errors may also come from the fact that we compare instantaneous satellite product to hourly mean values of the model.

## 4.3 Investigating AROME errors

We now identify potential sources of error in AROME that could explain the biases highlighted in section 3.3.

Wurtz et al. (2021) pointed out that in AROME, the ice and snow contents of the anvils of mesoscale convective systems 475 are too small. Their spatial extent is also too small, which they attributed to excessive snowfall velocities in AROME, resulting from a poor parametrization of the ice particle size distribution, an issue already reported by Taufour et al. (2018). Wurtz et al. (2023) has improved the treatment of snow in AROME, a correction that could help increase the snow mass and thus reduce the positive SWD bias for high clouds if snow is properly accounted for in the radiative code.



Another explanation for the positive SWD bias for high clouds could be that snow (which is one of the hydrometeors simulated by the microphysical scheme) is currently not taken into account in the AROME radiative code. Yet, on average over 2020, the total snow mass is 1.6 times that of cloud ice, meaning that a significant mass is neglected in the AROME radiative calculations. This is especially true for CR3, CR5, CR7 where the snow mass dominates both the cloud liquid and cloud ice masses (not shown). In practice, it is recommended to include snow in radiative calculations, as snow has a significant radiative impact on the SWD as shown in the IFS model (Li et al., 2014a) and CMIP simulations (Li et al., 2014b, 2022), at least in regions with high precipitation and/or convective activity. Applying the correction developed by Wurtz et al. (2023) and taking into account the radiative effect of snow could reduce the positive SWD bias for high clouds found in this study.

Apart from the treatment of snow and ice, SWD errors have often been attributed to liquid Water path (LWP) errors. Evaluation of LWP in AROME could provide information on LWP errors and help understand the SWD negative bias in overcast situations. However, evaluating the LWP at AROME resolution over a large domain and a full year is challenging since geostationary satellite LWP products, such as the NWC SAF product, remain very limited because they rely on passive instruments. To investigate SWD errors due to LWP errors, a case study could be conducted, such as Ahlgrimm and Forbes (2012) who identified LWP errors using microwave radiometer measurements that could explain the SWD positive bias in overcast low cloud situations in the IFS model. For instance, observations from the SIRTA supersite on the Saclay plateau (Chiriaco et al., 2018) could be used to better separate the contributions of variables such as cloud fraction, liquid water path and cloud droplets effective radius. Polar-orbiting satellites with active measurements onboard (e.g. CloudSat and CALIPSO, Stephens et al., 2018) could also be used to analyze the contributions of LWP and IWP errors, but their spatial and temporal coverage is very limited for evaluating the performance of NWP models at hourly and kilometric resolutions.

Errors in the representation of the mixed phase were also highlighted by Forbes and Ahlgrimm (2014) in the IFS model and Barrett et al. (2017) in five operational NWP models including the IFS model, as well as Engdahl et al. (2020) in HARMONIE-AROME. They reported an underestimation of the supercooled liquid water content in boundary layer clouds. This would likely result in an overestimation of SWD (Hogan et al., 2003), which could explain the positive SWD bias for the clouds involved, namely CR2, CR4 and CR5. Although this error source is not consistent with the average negative SWD bias we found for CR4 in overcast conditions, it may account for some positive errors (see Fig. 12) for CR4 and participate in the positive bias for CR2 and CR5.

To conclude, several sources of error have been highlighted in the literature, some of which may contribute to the SWD errors reported in our study. Further investigations with more advanced observations should be used to overcome the limitations of the observations used in this study.

## 5  Conclusions

In this study, we performed a detailed evaluation of the 24-hour SWD forecasts of the French NWP model AROME at 1.3 km horizontal resolution, comparing a full year of hourly forecasts with *in situ* SWD measurements from the pyranometer network operated by Météo-France. A preliminary analysis showed that errors occur mainly when clouds are present in the model and in



the observations, while erroneously predicted and missed clouds contribute less to the overall errors, as these situations are less frequent. Missed clouds and erroneously predicted clouds correspond mainly to clouds with a low impact on the SWD. Errors in cloudy situations, which have a positive bias overall, can also result from errors in the simulated cloud fraction or in the cloud optical thickness. Since errors in the cloud fraction are difficult to evaluate over a large domain, we limited our study to overcast situations in the model, implying that the SWD overestimation cannot be attributed to an underestimation of the cloud fraction. We then quantified the SWD errors for different cloud regimes, corresponding to different cloud altitudes. In doing so, we found a systematic SWD negative bias for low clouds and a systematic SWD positive bias for high clouds, consistent throughout the year with an overall average close to zero. In addition to these systematic deviations, we found unsystematic errors with significant SDE for all cloud regimes, which is critical for the energy sector. Note that this study was based on only one year, and even if the results seem to be robust with similar SWD errors along the year it might be relevant to extend this study to several years. In addition to these systematic deviations, we found unsystematic errors with significant SDE for all cloud regimes, which is critical for the energy sector. A cross-comparison between cloud regimes and cloud types showed relatively good agreement between the model and observations, especially for low clouds, confirming that the negative bias of SWD is indeed related to low clouds and the positive bias of the SWD is indeed related to high clouds. We pointed out that the positive bias of ice clouds may be due to the failure to account for snow in the AROME radiation code, or to the too low snow content of ice clouds, issues that could be addressed in the future. Others sources of SWD errors have been mentioned such as LWP or mixed phase representation errors.

Our results also suggest that some clouds are not detected by the satellite, highlighting the need for more detailed cloud observations to go further in error assessment. In addition, efforts should be made to evaluate the cloud fraction at AROME resolution over a large domain to fully characterize the performance of AROME for SWD forecasts. This may be addressed using large networks of ceilometers or shortwave radiation measurements. To better understand which physical properties of clouds cause SWD errors, further evaluations that distinguish cloud fraction errors from errors in the vertical distributions of condensed water and cloud particle effective radius are needed, and should be based on instrumented sites. Although the study focused on AROME, the presented methodology can be applied to any NWP model, and the detailed evaluation can provide valuable physical information about the model performance, paving the way for future model improvement.

*Data availability.* The cloud satellite products developed by the NWC Eumetsat SAF are available on this site: https://www.icare.univ-lille.fr/ and the documentation on: http://www.nwcsaf.org. The satellite product from Copernicus Atmosphere Monitoring Service (CAMS) are available on: https://atmosphere.copernicus.eu/. The data are freely available for research purposes at this address: https://donneespubliques.meteofrance.fr/?fond=produit&id_produit=298&id_rubrique=32. Access to the AROME code can be requested via the ACCORD consortium webpage: http://www.umr-cnrm.fr/accord/. AROME forecast are available on: https://doi.org/10.5281/zenodo.7928622. As the data set used for this study is very large (800Go), only the shortwave downward radiation is available. The others parameters can be made available on request from the corresponding author.



545  *Author contributions.* M.-A.M., Q.L., and S.R designed the methodology. M.-A.M. performed the analysis and prepared the figures. M.-A.M wrote the manuscript with contributions from all co-authors.

*Competing interests.* The contact author has declared that none of the authors has any competing interests.

*Acknowledgements.* The research leading to this work was carried out as a part of the Smart4RES project (European Union's Horizon 2020, No. 864 337). The sole responsibility of this publication lies with the authors. The European Union is not responsible for any use that may
550  be made of the information contained therein.

We acknowledge Météo-France and the Occitanie Région for funding the PhD of Marie-Adèle Magnaldo. We also thank Yves-Marie Saint-Drenan, Yves Bouteloup, Yann Seity, Marie Cassas and Emmanuel Fontaine for the fruitful discussions and/or the technical help.



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




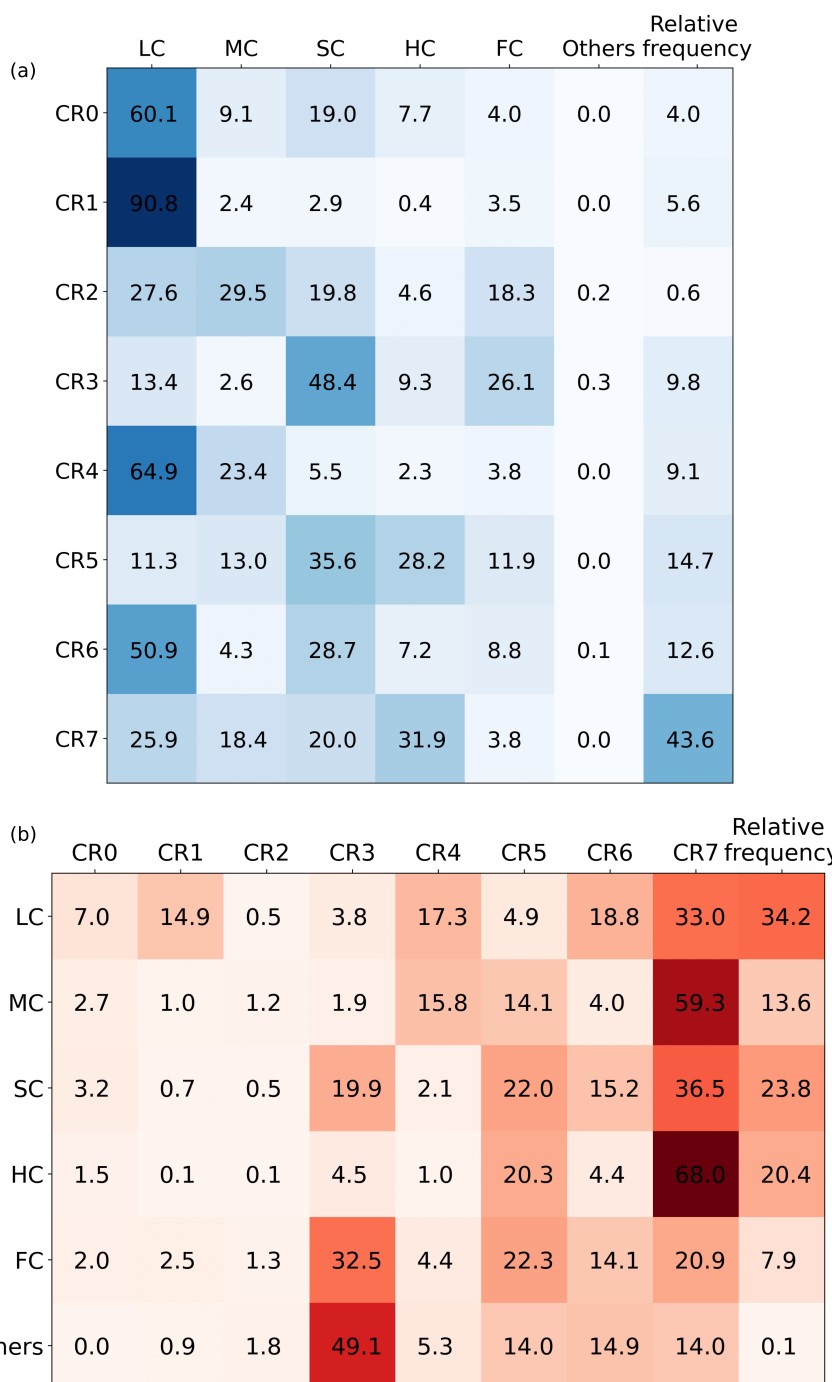

**Figure 14.** Cross-comparison of cloud regime/cloud type (in %) in hit cases. (a) For each CR, the relative frequency of each CT. The last column is the relative frequency of each CR for all hits in overcast situations. The sum of each row is equal to 1. (b) For each CT, the relative frequency for each CR. The last column is the relative frequency of each CT over all hits in overcast situations. The sum of each row is equal to 1.