# Peer review of "Evaluation of surface shortwave downward radiation forecasts by the numerical weather prediction model AROME"

_EGUsphere, 2023_

## Author Comment (AC1)

**Thank you very much for the positive feedback and all your relevant remarks and suggestions. We will try to answer all your questions in this document. The line numbers in our answers correspond to the numbers in the new corrected version.**

**Line 142: The results can be very sensitive to the choice of such thresholds. Did you investigate this? If so, please add a sentence on the reason of your choice in the article.**

There are several thresholds in our methodology: the SZA threshold (SZA < 84°), the cloud fraction threshold for cloud detection (CF > 2 %) and the neighborhood size (5x5 pixels). The following figures shows the Fig. 10a of the article along with equivalent figures with different values of these thresholds, in order to see how the results can be impacted.

[Figure]

*Figure 1: Monthly mean SWD bias (bar height, in W m$^2$ ) and relative frequency (bar width) for all modeled cloud regimes in 2020 and for different thresholds.*

The Figure 1 shows that our results are barely sensitive to the chosen thresholds, and, in particular, biases keep the same signs.

In practice, we have selected the cos(SZA)>0.1 (SZA<84.3°) threshold to keep most of the observations. Although at high SZA observations are less reliable, the corresponding SWD are low, so the impact on biaises and SDE is low. Note that Tuononen et al. (2018) and Ahlgrimm et al. (2012) also chose a poorly restrictive threshold of respectively 5 and 1 W m$^{-2}$.

In the same idea, we tried more than one value for neighborhood size (neighborhood size of 10 x 10 pixels and without neighborhood size (1 x 1 pixel ) by just selecting the closest point in the model), and likewise the systematic errors remain similar. Based on that, we chose the most relevant value 5 x 5 pixels, which is comparable to the satellite pixel size.

Regarding the cloud fraction threshold, the same value is chosen for cloud detection and cloud regime definition. Tuononen et al. (2018) chose a threshold for cloud detection of 5%, and Van Weverberg et al. (2018) chose a threshold of 0.5%. We have chosen an intermediate value of 2%, as in Ahlgrimm et al. (2012). Again, changing this threshold from 1 % and 10 % does not qualitatively change our results .

To be more accurate about the different threshold chosen, we did the following modifications :
L150, we added: "The neighborhood size is **chosen to be** comparable to the satellite pixel size."
L165, we removed: "The neighborhood size is comparable to the satellite pixel size."
L172, we added: "For each region, a 2 % threshold, **similar to the threshold value for cloud occurrence**, is set [...]"
L240, we added: "Note that several tests were performed by changing the neighborhood size **(1 pixel and 10 x 10 pixels)**, the cloud detection threshold **(1 %, 5 %, 10 %) and the SZA threshold (70°)**, which did not qualitively change these results."
L369, we added: "5 %"

**Line 174: Another threshold is introduced here. Again, as far as I understand, the sensitivity to this parameter has already been investigated. A short comment here would be helpful.**

See previous comment.

**Line 147-150: While neighborhood strategies are well-suited for verification purposes, this double penalty problem is exactly what PV power forecast have to struggle with. One could argue that this strategy is not suitable for solving the problems targeted in this study. An alternative strategy would be the usage of ensemble forecasts. As I understood from the following, you tried more than one value for neighborhood size and the results were not sensitive to this choice? Please clarify.**

We agree that PV producers expect the forecast to be accurate for the exact location of the PV plant.

However, the effective resolution of AROME is about 8 km. For AROME and for other NWP models, we have no reason to trust a forecast at a very fine scale (that of a PV plant), at a specific point and at a specific moment. By using NWP models, we accept that it can't be accurate at individual grid points. For more accurate very short term forecasts, other forecasting techniques are commonly use, such as machine learning approaches or all-sky imagers/satellite-based methods. It also depends on the size of the PV plants. For large ones, it's less important to know where exactly a cloud is locating as the production is aggregated over all PV panels. In any case, the double penalty does not prevent us from understanding SWD errors even if the methodology proposed does not solve this major limitation of NWP models.

We didn't look at ensemble forecast, mostly due to excessive data volume, and we already have a lot of information with operational deterministic forecasts. It is an interesting venue that should be further investigated. However handling ensemble forecasts would not be trivial and alternative metrics would be necessary to evaluate the performances of such ensemble forecasts.

We added in L144: "While such a neighborhood strategy may seem unsatisfactory to PV producers who are concerned about SWD at the exact location of the PV plant, these users should keep in mind that NWP forecasts are not expected to be accurate at individual grid points."

**Line 173 (Figure 2) : In the following, several terms are introduced that describe a combination of several cloud regimes. For example, "low clouds" which are cloud classes CR0, CR1, CR4. Another example is geometrically thin clouds which are CR1, CR2, CR3. A table here that summarizes these groupings of cloud regimes would be helpful. It would also help to avoid inconsistencies like for geometrically thick clouds which are CR4, CR5 and CR7 in line 286 and 438, but only CR5 and CR7 in line 464.**

We added a table (Table 1) that summarizes these groupings of cloud regimes.
The text was changed as follows at L457 to be more consistent and we added "high": "For optically thick **high** clouds that most likely correspond to CR5, CR7 and HC".

**Line 173 (Figure 2): Furthermore, a more descriptive naming of the cloud regimes might be beneficial for the readability of the manuscript.**

See previous comment.

**Line 177 It is not clear which special cloud regime is meant. CR7?**

This special cloud regime is just here to explain why the sum of relative frequencies is not totally equal to one in the figure 3a. As it was confusing, we changed the text in Line 177 : "Note that the sum of the relative frequencies of clear skies and all cloud regimes is not exactly equal to one in Fig. 3a. This is because there are situations, with very low relative frequencies, with a total cloud fraction larger than 2 % (implying that the scene is considered cloudy), but with a cloud fraction for each of the 3 vertical slices of the troposphere lower than 2 % (implying that it is not associated to any CR)."

**Line 195: This description is a bit misleading and on first reading it sounds like (SWD_mod/SWD_obs)/SWD_modclear. What you meant is probably SWD_mod/SWD_modclear for the model and SWD_obs/SWD_modclear for observations. One (or two) short formulas are always clearer than a description in words.**

We added an equation and changed the sentence Line 199 by:
"To begin with, we use the clear sky index (CSI), which is defined as : $\mathrm{CSI_{mod}} = \frac{\mathrm{SWD_{mod}}}{\mathrm{SWD_{clear,mod}}}$ ; $\mathrm{CSI_{obs}} = \frac{\mathrm{SWD_{obs}}}{\mathrm{SWD_{clear,mod}}}$,
with $\mathrm{SWD_{mod}}$ the SWD in the model, $\mathrm{SWD_{obs}}$ the SWD in the observations, and $\mathrm{SWD_{clear,mod}}$ the theoretical SWD under clear sky conditions from the model. The histograms of CSI are used to […]"

**Line 201: This clear-sky underestimation of SWD is not surprising. There is a decreasing aerosol trend in Central Europe in the last decades. Older climatologies (like Tegen) are thus overestimating present-day aerosol loadings in Central Europe.**

It is a good point. We added a reference and a sentence at line 207:
"In the observations the CSI can greatly exceed 1, as reported by Nielsen et al. (2018). This can be explained by an underestimation of the model clear sky SWD, which can be due to a reduction in aerosol emissions over the last decades (Wild et al., 2009) while the aerosol climatology used in AROME is older (Tegen et al., 1997) and thus overestimates present-day aerosol loadings. These values exceeding 1 can also be due to cloud enhancement effects (e.g. Gueymard et al., 2017), which occur under broken cloud conditions."

**Line 247: As mentioned before, there is also a positive bias in Tegen AOD for Central Europe.**

Actually we found an overall positive bias of SWD, which can not be attributed to this positive AOD bias. Hence we don't mention this in this section.

**Line 416: "...and the SWD higher (15%)" Do you mean SD?**

Yes, corrected.

**Line 428-431: I did not fully understand this paragraph. It starts with the description of the second pair of values, the conclusion seems to be rather suitable for the third pair of values which is not described anywhere else.**

Finally, we removed the Fig.13 and changed the text of this section. Please refer to our reply to Anonymous Referee #2 to check these modifications.

**Line 446: For this section where you verify the satellite-based cloud classification of your approach, SYNOP data on high, mid and low cloud fraction might be helpful as an independent dataset looking from a different perspective. Have you considered this?**

We believe that you refer to telemeters data or human observations. We indeed consider both types of observations, however the data are not available at all pyranometer stations used in this article. In addition, for human observations, cloud altitude can be very difficult to estimate, hence the information is highly subjective and not adapted for a quantitative study. Furthermore, cloud regime evaluation wasn't our primary objective. Nevertheless, in a companion study (that may be published in the future), we are comparing modeled cloud regimes with data from a highly instrumented site where lidar and radar observations are available.

**Line 481: Did you separate between low and high clouds here? For low clouds, the consideration of snow might even further pronounce the already existing bias.**

No, we didn't separate low and high clouds here.
We agree that taking snow into account could accentuate the existing bias. However, in an NWP model, there would be no reason to account for snow in high clouds only, and not low clouds.
The figures below show the profiles of the monthly snow mass normalized by the cloud hydrometeors mass for each cloud regime, in February 2020 (a) and in August 2020 (b) in AROME forecasts.

[Figure]

*Figure 2 : Profiles of mean $q_{snow}/(q_{ice}+q_{liq})$ (where $q_{snow}$ is the mass of snow, $q_{ice}$ is the mass of cloud ice and $q_{liq}$ the mass of cloud droplets) in (a) February 2020 and (b) August 2020 in the model AROME and for each simulated cloud regime.*

It shows, indeed, that taking snow into account could accentuate the existing SWD bias, in particular for CR4 in Winter. To further investigate this, we run simulations using a more recent version of AROME without (named: New version of AROME) and with accounting for snow (named: +Snow) for only two months (February and August 2020). The results are shown in Fig. 3, in terms of bias and SDE for each cloud regime in overcast conditions.

[Figure]

*Figure 3: For a new version of AROME and for the simulation +Snow: in the first line, monthly mean SWD bias (bar height, in W $m^2$ ) and relative frequency (bar width) for all modeled cloud regimes for February and August 2020. In the second line, monthly SDE (in W $m^2$ ) for all modeled cloud regimes.*

It confirms that the SWD bias of CR4 is deteriorated when snow is taken into account in the radiation scheme. However, the overall error is reduced even if the bias for some cloud regimes is accentuated, as the bias and SDE are greatly improved for some cloud regimes. In any case, this issue needs further investigation. For information, the snow is already taken into account in the radiative scheme of the IFS model, as its radiative impact is not negligible.

We added at line 479: "Note however that it could also deteriorate the bias for already too opaque low clouds (e.g. CR4 in winter)."

**References:**

Ahlgrimm, M. and Forbes, R.: The Impact of Low Clouds on Surface Shortwave Radiation in the ECMWF Model, Monthly Weather Review, 140, 3783–3794, https://doi.org/10.1175/mwr-d-11-00316.1, 2012.

Gueymard, C. A.: Cloud and albedo enhancement impacts on solar irradiance using high-frequency measurements from thermopile and photodiode radiometers. Part 1: Impacts on global horizontal irradiance, Solar Energy, 153, 755–765, 2017.

Nielsen, K. P. and Gleeson, E.: Using shortwave radiation to evaluate the HARMONIE-AROME weather model, Atmosphere, 9, 163, 2018.

Tegen, I., Hollrig, P., Chin, M., Fung, I., Jacob, D., and Penner, J.: Contribution of different aerosol species to the global aerosol extinction optical thickness: Estimates from model results, Journal of Geophysical Research: Atmospheres, 102, 23 895–23 915, https://doi.org/DOI:10.1029/97JD01864, 1997.

Tuononen: Evaluating solar radiation forecast uncertainty, Atmospheric Chemistry and Physics, 19, 1985–2000, https://doi.org/10.5194/acp-19-1985-2019, 2019.

Weverberg, K. V., Morcrette, C. J., Petch, J., Klein, S. A., Ma, H.-Y., Zhang, C., Xie, S., Tang, Q., Gustafson, W. I., Qian, Y., Berg, L. K., Liu, Y., Huang, M., Ahlgrimm, M., Forbes, R., Bazile, E., Roehrig, R., Cole, J., Merryfield, W., Lee, W.-S., Cheruy, F., Mellul, L., Wang, Y.-C., Johnson, K., and Thieman, M. M.: CAUSES: Attribution of Surface Radiation Biases in NWP and Climate Models near the U.S. Southern Great Plains, Journal of Geophysical Research: Atmospheres, 123, 3612–3644, https://doi.org/https://doi.org/10.1002/2017JD027188, 2018.

Wild, M.: Global dimming and brightening: A review, Journal of Geophysical Research: Atmospheres, 114, 2009.

---

## Author Comment (AC2)

**Thank you very much for the positive feedback and all your relevant remarks and suggestions. We will try to answer all your questions in this document. The line numbers in our answers correspond to the numbers in the new corrected version.**

**Abstract: While SWD is defined on the first line, it might be helpful for clarity to mention somewhere in the abstract body how the sign of the bias is defined (e.g. where a positive bias is first mentioned, one might add "i.e. too much shortwave radiation reaching the ground" or similar).**

Yes, we modified the sentence: "The 2020 bias is positive, with a value of 18 W m$^2$, **meaning that AROME overestimates the SWD**. The root-mean-square-error [...]".

**L90: Single sites are not necessarily unrepresentative, particularly if long time series are considered that cover a variety of cloud regimes. I find the wording a bit strong here.**

Yes, we agree. We modified the sentence: "**Hence, it should not be limited to a single supersite, because such a site could be unrepresentative of the whole domain.** The method should also allow to go..."

**L137: I find the reference to Table 1 here a bit confusing. The text talks about 15 cloud types, yet the table lists only 6. The merging of the cloud types is discussed later in the text. Maybe instead of referencing Table1 in line 137, the text should refer the reader to section 2.2.3 where the table is explained (and first reference the table in that section).**

We modified the text as follows L134: "The cloud type is a classification among 15 different classes: 4 corresponding to cloud free scenes **and** 11 corresponding to clouds scenes (**see Section 2.2.3**)."

**L145: What is the sensitivity to size of neighbourhood, or choosing the nearest-neighbour instead of using a neighbourhood approach?**

The Anonymous Referee 1 also asked this question. We invite you to read our response to this reviewer.

**L188: I don't really see a more pronounced seasonal cycle. Does that statement apply to the low cloud category only, or to all cloud? If it applies to only low clouds, it is not easy to make out the seasonal cycle of CR0+CR1+CR4 in Fig. 3 since the categories are not grouped together. Please clarify.**

We agree that the comparison of the seasonal cycle of both figures is not easy. Hence we modified the text to focus only on the seasonal cycles of cloud types, where fractional clouds (FC) and low clouds (LC) exhibit quite clear seasonal cycles.  In contrast low clouds in the model do not show such a seasonal cycle (see figure below where we put side by side CR0, CR1, CR4).
L188: "**the relative frequency of observed fractional clouds is higher in summer than in winter, while low clouds are overall less frequent in spring and summer,  highlighting seasonal cycles in the observations that do not have obvious equivalents in the model**.".

[Figure]

*Figure 1 : Monthly relative frequency for each (a) cloud regime in AROME and cleasr ckies (CS), (b) cloud type in the satellite images and clear skies (CS), over 2020 for the pixels including the piranometers.*

**L208: I'm not that familiar with various configurations of AROME. How does HARMONIE-AROME differ from the AROME version presented in the manuscript? Maybe a half-sentence would be useful here, e.g. stating**

**that HARMONIE-AROME uses the same microphysics and radiation schemes (if that is the case), to indicate that the results from the cited study apply.**

We changed the sentence and added indications: "Interestingly the CSI distributions are similar to those reported by Nielsen et al. (2018) with HARMONIE-AROME **(Bengtsson et al., 2017)**. **Although these models share the same code, the operational configurations rely on different sets of parameterizations.**"

**Fig 12 does not add much to the discussion. I think it (and the few sentences discussing it) could probably be left out.**

We agree that this figure is partly redundant with the previous ones. However, we think that it shows an important aspect we woud like to insist on, which is the contributions of both positive and negative errors for all cloud regimes. This cannot be seen on the other figures and is now more clearly highlighted in the text :
L384: "**To further investigate the SDE associated to each cloud regime**, Fig.12 shows the distribution of SWD errors for all AROME cloud regimes. For each CR the SDE is high compared to the mean flux, in particular for CR2, CR4, CR5 and CR6. **Interestingly, it shows that for all cloud regimes the mean biases result from both positive and negative contributions, indicating that multiple sources of errors are involved**. The same is obtained for the distribution of SWD errors for observed cloud type in the satellite classification (not shown). **This suggests that improving the mean bias of individual cloud regimes would not necessarily imply much better forecasts. It also implies** that more detailed observations are needed to better understand these errors and their sources."

**Section 4.1.2: I found this section most difficult to follow, and also somewhat repetitive. It seems to mainly confirm conclusions that were already drawn previously from analyzing SWD and SD in cloud classes/types. That does not really surprise me, since I would expect the informational content found in SWD, SD and clear sky index (extensively discussed in sections 3.3) to be the same as found in the transmittance and its standard deviation.**

**E.g. L185: "some optically thin clouds are not detected by NWC SAF product, which is a known caveat of passive sensors" – L418: "clouds may be present but not detected"**

**Similarly, we have already seen that the contribution to the SWD bias from false alarm and missed cases is relatively small (corresponding to the conclusions on Line 430 and 434).**

**I would like this section to be more concise (what new information does the transmittance perspective contribute, that hasn't been seen the section 3.3 previously?), or maybe some of the additional information could be wrapped in with the discussions throughout section 3.3, eliminating 4.1.2.**

We agree. For the readability of the article, we decided to remove Fig.13 and the text refering to it. As you point out, it seems clear that some clouds are undetected by satellite product. We kept section 4.1.2 (although much shortened) and the paragraph related to the impact of this non-detection of clouds on our results and to a possible solution of this issue. In order to keep the information about the fact that missed clouds and undetected simulated clouds have on average less impact on the SWD, we changed the text as follow:

We added at L297 : "**Note that the annual mean SWD in the model for the false alarm cases (436 W m$^2$) is much higher than the annual mean SWD in the model for the hit cases (296 W m$^2$), suggesting that undetected simulated clouds have on average less impact on the SWD than actually observed clouds.**"

We added at L321: "**Note that the annual mean SWD in the observations for miss cases (409 W m$^2$) is much higher than for hit cases (272 W m$^2$), suggesting that the clouds missed by the model have, on average, a small impact on the observed SWD.**"

We added L422: "**In addition, the temporal variability of SWD under clear sky is much larger in the observations than in the model (not shown).**"

**L498: I am not sure how relevant lacking supercooled liquid is likely to be over France. The greatest impact on radiation is found in cold regions where models erroneously produce ice-only clouds. I'd expect most LC in France to contain liquid water under most conditions, except maybe for some winter conditions. Nevertheless AROME is used elsewhere, and the Scandinavian countries could certainly benefit from an improved representation of supercooled liquid.**

Underestimation of supercooled liquid occurs also over France and causes problems as wind turbine blade and aircraft icing. Studies are carried out at Météo-France on this topic. This underestimation is mainly related to mid-altitude clouds in France, and not much to low clouds as in cold regions. We can't say to what extent this underestimation has a strong radiative impact, but it certainly exists.

**Technical comments:**

**L25: should this link be in the text, or in a reference? Not sure what the journal's style guide suggests, but a long link in the text disrupts the flow a bit.**

We agree. We used footnotes for the two links in the article for a better readability.

**L38/39: wording:" optimal reserves optimizing storage."**
Corrected in "optimizing storage".

**L49: typo: short-terme, should be term, also: satellite should not be capitalised**
Corrected.

**L90: typo: word "not" is used twice**
Corrected.

**L218: Wording suggestion: Use "In contrast, "instead of "on the contrary" (applies to several places throughout the text)**
Corrected.

**L249: wording: "local and punctual" – suggestion: just use "localised". "punctual" means "being on time". Or if you are referring to time, then maybe "temporary" or "short-lived" might be better.**
Corrected.

**L294: The year is missing in reference Antoine et al. – looks like it is not published yet. In this case, maybe there should be a further comment behind the name, e.g. "in preprint", "under review" or similar (not sure what the journal prefers).**
Indeed, this article was not yet published, but now it is. Corrected.

**Fig 11: Labels of the cloud type (CRx) in each panel would make it easier to follow the discussion in the text**
Yes, we changed the figure.

**L393: remove parentheses around reference to Lucas-Picher et al.**
Corrected.

**L428: put Ackermann reference in parentheses**
Corrected.

**L520 and following: sentences are repeated**
Corrected.

**L542: Should the units be Gb, not Go?**
Yes, corrected.